# Microplastic Removal in Wastewater Treatment Plants (WWTPs) by Natural Coagulation: A Literature Review

**DOI:** 10.3390/toxics12010012

**Published:** 2023-12-22

**Authors:** Taskeen Reza, Zahratul Huda Mohamad Riza, Siti Rozaimah Sheikh Abdullah, Hassimi Abu Hasan, Nur ‘Izzati Ismail, Ahmad Razi Othman

**Affiliations:** 1Department of Chemical and Process Engineering, Faculty of Engineering and Built Environment, Universiti Kebangsaan, Bandar Baru Bangi 43600, Selangor, Malaysia; p126930@siswa.ukm.edu.my (T.R.); p113185@siswa.ukm.edu.my (Z.H.M.R.); rozaimah@ukm.edu.my (S.R.S.A.); hassimi@ukm.edu.my (H.A.H.); nurezatyismail@ukm.edu.my (N.‘I.I.); 2Research Centre for Sustainable Process Technology, Faculty of Engineering and Built Environment, Universiti Kebangsaan, Bandar Baru Bangi 43600, Selangor, Malaysia

**Keywords:** wastewater treatment plant, coagulation, microplastics, marine environment, Southeast Asia

## Abstract

Urban industrialization has caused a ubiquity of microplastics in the environment. A large percentage of plastic waste originated from Southeast Asian countries. Microplastics arising from the primary sources of personal care items and industrial uses and the fragmentation of larger plastics have recently garnered attention due to their ubiquity. Due to the rising level of plastic waste in the environment, the bioaccumulation and biomagnification of plastics threaten aquatic and human life. Wastewater treatment plant (WWTP) effluents are one of the major sources of these plastic fragments. WWTPs in Southeast Asia contribute largely to microplastic pollution in the marine environment, and thus, further technological improvements are required to ensure the complete and efficient removal of microplastics. Coagulation is a significant process in removing microplastics, and natural coagulants are far superior to their chemical equivalents due to their non-toxicity and cost-effectiveness. A focused literature search was conducted on journal repository platforms, mainly ScienceDirect and Elsevier, and on scientific databases such as Google Scholar using the keywords Wastewater Treatment Plant, Coagulation, Microplastics, Marine Environment and Southeast Asia. The contents and results of numerous papers and research articles were reviewed, and the relevant papers were selected. The relevant findings and research data are summarized in this paper. The paper reviews (1) natural coagulants for microplastic removal and their effectiveness in removing microplastics and (2) the potential use of natural coagulants in Southeast Asian wastewater treatment plants as the abundance of natural materials readily available in the region makes it a feasible option for microplastic removal.

## 1. Introduction

The commercial use of plastics began to rise in the 1950s. The world began relying on plastic products due to their versatility, durability, degradation resistance and low cost [1]. The increasing demand for plastics, however, came with the consequence of plastic pollution in the environment, which could have undesirable effects on nature and the living beings in it [2]. According to the Organisation for Economic Co-operation and Development (OECD), plastic consumption has quadrupled over the past 30 years, with global production reaching 460 million tons in 2019. The plastic waste generated by consumption has also doubled to 353 million tons between 2000 and 2019 [3]. Microplastics are synthetic plastic materials with a diameter of less than 5 mm [4]. Even though microplastics were first mentioned in the 1970s [5], it was not until 2004 that these substances gained public and scientific awareness. Microplastics are generally classified into two categories: (1) primary microplastics and (2) secondary microplastics [6]. Currently, microplastic pollution in the environment is ubiquitous and found in oceans, coral reefs and polar regions; it mostly originates from land, with wastewater treatment plant effluents being a significant source of this pollutant [7]. The small size of microplastics is hazardous to marine organisms as they can be mistaken for food and consumed by these organisms [8]. The microplastics from marine organisms can be transferred to human beings through consumption. Furthermore, microplastics have been found in sea salt and drinking water, which could also adversely affect public health [9]. Asia is the largest continent in the world and, thus, the largest contributor to marine plastic pollution. Approximately 81% of marine plastic is from Asian countries. Southeast Asian countries such as Indonesia and the Philippines were found to be the prominent contenders responsible for marine plastic pollution [10]. This scenario is mainly due to the incompetence of wastewater treatment plants in Southeast Asia in removing microplastics [10]. One of the significant barriers to wastewater management in Southeast Asian countries is the cost.

It is crucial to mitigate the plastic pollution from wastewater treatment plants as they release a large percentage to the environment. An estimated 3.85 × 10^16^ microplastics per year are released from wastewater effluents [11]. Existing primary and secondary treatment processes can remove approximately 66% of the microplastics in the influent [12,13]. Coagulation is the process of removing contamination in suspended particle and colloidal forms by destabilizing and aggregating the particles into large flocs. The aggregates then settle and can be removed from water using a solid–liquid separation method [14]. Coagulation is a simple and cost-effective technology used in water treatment plants. In the wake of sustainable development, research on natural coagulants as replacements for chemical coagulants has increased. Natural coagulants are renewable, biodegradable, non-toxic and cheap, making them more attractive than chemical coagulants [15]. In recent studies, chemical and natural coagulants could effectively remove microplastics in wastewater streams. However, the research on natural coagulants for microplastic removal is limited, and most research focuses on turbidity and COD removal. Despite the limited research, natural coagulants have proven efficient and can help mitigate the microplastic problem. Coagulation using natural coagulants is a sustainable and suitable solution for the microplastic problem in wastewater effluent. The mechanism involved in natural coagulation is assumed to be a combination of two or more mechanisms. Charge neutralization and bridging are the most probable mechanisms of action of natural coagulants. Southeast Asia is abundant in natural resources such as bananas and Moringa oleifera, which have proven to be efficient in removing plastics [16,17]. Improving the extraction and purification of these coagulants can enhance the removal efficiency, but further studies are required for these improvements. There is limited research on microplastics in wastewater effluent and natural coagulation in Southeast Asian countries. Filling these research gaps will help mitigate the microplastic problem in this region.

This review aims to (1) understand and evaluate the process of coagulation using natural coagulants to remove microplastics in wastewater effluents, (2) identify the feasibility of natural coagulants in Southeast Asian countries and (3) identify the future research pathways regarding microplastic removal using coagulation.

## 2. Microplastics from Wastewater Treatment Plants (WWTPs)

Almost 98% of the microplastics in the marine environment are generated from land activities, with road runoff being the primary source, followed by treated effluents from wastewater treatment plants (WWTPs) [18]. Primary microplastics are directly introduced in wastewater effluent streams, and the washing process of synthetic textiles is considered the major source of these primary microplastics in the oceans. The minute size of the plastic particles allows them to traverse wastewater treatment plants (WWTPs) and enter marine environments [19]. Wastewater treatment plants (WWTPs) are a significant source of microplastics in the environment. Microplastics can enter WWTPs in a variety of ways, such as sewage and stormwater runoff, and are discharged into the environment along with the treated wastewater. Primary microplastics from personal care items, the fibers from textiles during washing in domestic wastewater [20], and industrial effluents containing plastic fragments used in molding and other processes are major microplastics in plants. The wet sedimentation process washes off the tiny microplastic dust particles in the atmosphere resulting mainly from the wear and tear of tires, and road markings are carried to the treatment plants through stormwater runoff [21]. Plastic wastes undergo mechanical degradation, leading to fragmentation due to extreme environmental conditions in landfills. The leachate discharge carries the plastic debris to WWTPs [22]. In a study by He et al., it was found that all 12 leachate samples investigated showed the presence of microplastics [22].

Identifying the shapes of the microplastics present in the wastewater is necessary as this helps with the implementation of removal technology in a WWTP. The most common shapes in wastewater are fibers, pellets, fragments and films [23]. Fibers are the most dominant shape, accounting for nearly 52.7% of the microplastics present in wastewater. This can be due to the enormous quantity of fibers discharged in domestic washing discharges [24].

Although it is challenging to come up with an actual amount of microplastics released by WWTPs, it is well established that most of the microplastics in the marine environment come from wastewater [25]. Murphy et al. estimated that a WWTP serving a population of 650,000 could release up to 65 million microplastics into the marine environment daily [26]. It was shown statistically that approximately 8 trillion were entering the marine environment daily through wastewater systems [27]. Treated and untreated wastewater effluent can contribute possibly 3.85 × 10^16^ microplastics per year [11]. Europe alone was estimated to release 520,000 tons of plastic in wastewater effluent streams [28]. In a study conducted in Australian WWTPs, it was found that approximately 22.1 × 10^6^ to 133 × 10^6^ microplastics enter the environment per day through wastewater effluents [29].

In Asia, most WWTPs are unable to completely remove the microplastics in the influent, with a high percentage of microplastics remaining in the effluent streams even after the treatment process; thus, wastewater streams are one of the largest contributors of microplastic pollution in the environment. Even though Asia is the largest contender in the microplastic problem, no cumulative data on the amount of microplastics entering the marine environment through WWTPs could be found [30,31]. Despite Southeast Asian countries being among the top contributors of microplastics in the oceans, there are limited data available on the microplastics from wastewater. In a WWTP in Thailand, the final effluent, on average, contained 10.67 particles of microplastics per liter of wastewater [32]. In another study conducted in Thailand on three WWTPs, it was found that, on average, two pieces of microplastics were present per liter of wastewater effluent [33]. In Malaysia, a study conducted on the Semenyih River showed that approximately 7.47 microplastic particles are released per liter of wastewater from WWTPs [34]. In Vietnam, the density of microplastics in wastewater effluent was between 0.684 and 2.107 g/L [35]. In another study conducted in Vietnam, the effluent contained between 140 and 813 microplastic items per m^3^ [36]. In Surabaya, Indonesia, treated water contained an average of 10.4 plastic particles/L [37]. Few studies have proven the persistence of microplastics in wastewater even after being treated.

### 2.1. Microplastic Removal in WWTPs

Current wastewater treatment plants are not intended to remove the microplastics that appear with the waste. As microplastics are an emerging pollutant, specific treatment plants have yet to be created to eliminate them. A removal efficiency of more than 88% could be reached with secondary treatment [24] with the efficiency increasing to 99.9% with tertiary treatment [21]. The fundamental design of municipal WWTPs around the world is relatively the same, with Figure 1 showing the standard processes included in the primary, secondary and tertiary treatment steps.

In the primary treatment, microplastics larger than 1000 µm can be removed [38]. Primary treatment can also remove fiber microplastics as the fibers are trapped due to flocculation and settling [24]. Conventional primary treatment was able to remove approximately 65% of the microplastics in the influent [12]. In a study conducted by Bayo et al., a removal efficiency of 74% was achieved in the primary stage of the WWTP [39]. In one study conducted, primary treatment including coagulation was able to remove 98% of microplastics, and in another study, the removal efficiency reached up to 95.3% [40,41]. In a WWTP without the coagulation process, the mean removal efficiency after treatment was found to be 72% [42]. Microplastics can become entrapped within the aeration tank during the secondary treatment of microbial polymers or sludge flocs. The microplastic removal efficiency was 67% in the activated sludge process [13]. Microplastics with a particle size of more than 500 µm were found to be absent after secondary treatment [43]. Tertiary treatment can reduce the amount of microplastics in the influents to 0.2 to 2%. Talvitie et al. conducted a study to compare the removal efficiency of different tertiary treatment technologies and found the membrane bioreactor to be the most efficient, with an efficiency of 99.9% [44]. Studies conducted on WWTPs have shown that the majority of the microplastics that remain after the tertiary treatment processes are fibers. This could be due to the fibers being able to traverse the membranes longitudinally [45].

In Wuhan, China, a WWTP could remove 65% of the microplastic present [46], and in Sydney, Australia, the overall removal was 66% [13]. Compared to that, a wastewater treatment plant in Vancouver, Canada, removed 91.7% of microplastics [47], while another plant in Finland could achieve 99.9% microplastic removal [44]. The difference in the efficiency of removing microplastics is due to the technologies implemented in WWTPs in different regions. Most WWTPs in Asia and Australia use conventional treatment systems, which include screening followed by primary sedimentation and secondary treatment. WWTPs in Europe and certain North American countries have equipped advanced treatment methods, which include membrane bioreactors and dissolved air floatation.

The size, morphology and type of polymer influence the amount of microplastic removed in wastewater treatment plants. Primary treatment can remove bigger microplastics as well as fibers efficiently. In secondary treatment, fragments have a high efficiency of removal as they agglomerate and are ingested by the activated sludge [48]. Specific plastic shapes, such as pellets, were removed easily during tertiary treatment. The tertiary treatment process is also able to remove microplastics with tiny particle sizes [23].

### 2.2. Advanced Removal Technologies

Although microplastics in wastewater can be removed during the primary, secondary, and tertiary segments, none of the processes involved is specifically devised to remove microplastics. This causes a significant amount of microplastics to remain in the WWTP effluent, which releases these microplastics into the environment. Most microplastics are contained in sewage sludge and can be distributed through sludge land application. Advanced technologies, such as rapid sand filtration, the sol–gel method, electrocoagulation and photocatalytic degradation, are some approaches proposed for removing microplastics in WWTPs [49]. Most technologies are designed as add-on technologies for the existing secondary and tertiary treatment facilities. Rapid sand filtration is proposed as a tertiary treatment substitute to eliminate microplastics. Thus, a pre-treatment with techniques such as sedimentation and coagulation is necessary before the application of this process [50]. The electrocoagulation process causes the separation of microplastics through flotation by dissolving sacrificial anodes to free the coagulant precursors, which causes electrolysis to occur at the cathode and is depicted in Figure 2 [51].

In studies conducted on electrocoagulation, the process was able to remove 99.2% of microbeads from wastewater at a pH of 7.5 [52] and remove more than 80% of the COD and color from industrial wastewater [53]. The process was also able to remove 98.6% of microplastics from wastewater in another study [54]. CuFeO_2_@EP photocatalysts were able to degrade 99% of methylene blue dye [55]. Several studies were conducted to produce photocatalysts using the sol–gel method for the treatment of wastewater. In one study, floating Bi–N–TiO_2_ photocatalysts made using the sol–gel method were able to degrade 83.8% of diesel oil at a temperature of 550 °C [56]. Ni–N–TiO_2_ photocatalysts produced using the sol–gel method also showed high efficiency in the degradation of diesel oil. At a temperature of 550 °C and a degradation time of 300 min, they were able to degrade 95.9% of oil [57], whereas in another study, only 63.0% of oil was degraded [58]. Rapid sand filtration was also able to remove 75.49% of the microplastics present in wastewater influent with an average microplastic concentration of 4.40 ± 1.01 MP/L [59]. A few studies show that these technologies can be considered for the microplastic removal process in WWTPs as they have shown excellent results. They are, however, complex, and there are substantial research gaps in the optimization of these processes, which need to be filled before the implementation of the techniques on large scale. Extended study and research is required to understand the processes better as they are still at a preliminary stage, and it will be a long time before any of the technologies can be implemented in the existing WWTPs. Although, these technologies show excellent results, they are still in the development stage compared to coagulation, which is implemented by most WWTPs around the world. The improvement of the coagulation process for microplastic removal is therefore a feasible alternative to these specific advanced technologies for microplastic removal.

### 2.3. Wastewater Management in Southeast Asia

Most WWTPs in Asia cannot remove the microplastics in wastewater influents. This is a major issue as it has been noted that Asia is the primary source of plastic waste pollution. The wastewater facilities in Southeast Asia are currently dominated by decentralized wastewater technologies (DWTs), with cluster wastewater technologies (CCTs) in the urban areas. Only a few centralized wastewater technologies (CWTs) were observed [60]. The different systems can be defined based on the treatment capacity and the proximity to the wastewater source. DWT systems’ approach is treating the wastewater at or near the source. DWTs have a lower capacity of approximately 5000 person equivalents, whereas the capacity of CWTs is twofold that of DWTs [61]. CCTs are usually classified under CWTs as both systems treat wastewater from multiple households, whereas DWTs can only treat a single household. The major difference between CCTs and CWTs is the size of the facility, and CWT systems are considered large-scale CCT systems. CWTs collect wastewater from multiple households, and the wastewater is then carried to an end-of-pipe treatment facility. DWTs, on the other hand, treat the wastewater within the building with minimal collection. The facilities are minimal, and the standard technologies used are aerobic and anaerobic digestion, composting, sand/soil filtration and wetlands [62]. Both CCTs and CWTs are formed by a vast sewer network that carries the wastewater from sources to the treatment plant. With the growing population and urbanization in the countries considered, CWTs are considered the preferred treatment facility. However, most CWT facilities have deteriorated over time and cannot perform at their full capacity. A major issue with DWT systems is that they are unable to comply with the limits of environmental discharges [63]. In many cities such as Bangkok [64] and Kuala Lumpur in Southeast Asian countries, the CWTs have poor effluent quality due to overcapacity.

## 3. Coagulation

There has been recent research on advanced technological methods to mitigate the number of microplastics in wastewater effluent. However, the specific treatment processes still need to be applied on a full scale to any wastewater treatment plant. Furthermore, implementing the technologies in the existing wastewater treatment plants could increase the plants’ capital and operational costs. An economical solution to the cost problem would be to tune the operational parameters of the existing treatment processes to increase the efficiency of removing microplastics. Improving the flocculation and coagulation process could be essential in removing microplastics [24]. In a study conducted by Ma et al., it was observed that an aluminum-based coagulant showed improved efficiency in the removal of microplastics, which implies the possibility of improving the process of coagulation in wastewater treatment plants [65].

The coagulation process, as shown in Figure 3, consists of merging small particles into larger aggregates or flocs, followed by the adsorption of dissolved organic matter into the flocs. The flocs are removed as impurities in subsequent solid–liquid separation processes [14]. Coagulation is an important operation in wastewater treatment plants and for sludge dewatering in industries such as the pharmaceutical, pulp and paper processing and metalworking industries [66]. Coagulation is a commonly used treatment method due to it being cost-friendly and easy to operate [67].

Coagulation could be used in WWTPs to remove microplastics. The addition of a chemical or natural coagulant neutralizes the charge on the surface of the microplastics, causing them to clump together and form flocs. The flocs can then be removed using a solid–liquid separation method. In studies conducted, existing coagulation had a high removal efficiency of 47 to 82% [68]. Another study stated that the coagulation process in the tertiary treatment sector had a higher average removal efficiency of 64.4% compared to that in the primary (60.6%) and secondary (60.6%) treatment sectors [69]. In an investigation conducted in Tianjin, China, coagulation removed 76.4% of the microplastics present in wastewater [70]. In another study conducted in Hong Kong, coagulation removed 78.2% of suspended microplastics [71]. It is worth noting that the removal efficiency of coagulation is highly dependent on the characteristics of microplastics. In general, it is found that coagulation can remove larger sizes of microplastics [37] and fibers more efficiently [70]. Magnetic coagulation is also more effective in removing microplastics from wastewater.

### 3.1. Mechanism

The interaction of different coagulants is varied via a broad range of mechanisms and kinetic processes. In general, the kinetics of coagulation is described as contact between the coagulant and suspended colloids by absorption through electrostatic interactions. The conformation of adsorbed polymers is rearranged, resulting in aggregation in the suspended particles, which causes them to form large flocs [67]. The mechanism involved in the coagulation process largely depends on the type of coagulant used as well as the properties of suspended particles.

In general, the coagulation mechanisms can be sorted into four types: (1) Simple charge neutralization is the neutralization of the charges present in the colloidal surface. A decline in the electrostatic repulsion to a minimum value causes the particles to aggregate and form large flocs. (2) In charge patching, heterogeneous charges on the colloids are unevenly distributed, which generates electrostatic attraction in the particles. The non-zero value of the zeta potential at the optimal dose forms a flocculation window. The electrostatic attraction leads to the eventual aggregation of the particles, forming large flocs. (3) Bridging usually occurs when the molecular weight of the coagulants is high. The long-chain coagulants connect the finer flocs to accumulate into a large one. (4) The sweeping mechanism is used by inorganic coagulants. Hydroxide precipitates are formed as a fine colloidal dispersion. Further aggregation produces hydroxide flocs [67,72,73,74]. The detailed mechanism method is shown in Figure 4.

The coagulation mechanism for removing microplastics can be assumed to be a combination of two or more coagulation mechanisms. The properties of the wastewater also play a significant role in the mechanism involved in the process. Charge neutralization plays a role as the suspended microplastic particles have a negative surface charge [75]. Adding a coagulant to the wastewater will result in the neutralization of the surface charge of the microplastics. For coagulants with large molecular weights, the bridging mechanism also plays a role in removing microplastics. Bridging is exerted by the coagulants as they link with microplastics that have not reached complete destabilization through electrostatic gravitational forces and Van der Waals’ forces [67,76].

### 3.2. Factors Affecting Coagulation

Several factors and operational conditions can affect the process of coagulation. It is important to understand the general and specific factors affecting the process to help with the optimization. The principal operating conditions that affect coagulation are pH and temperature. Different coagulants tend to have different pH dependencies. The initial pH level can affect the surface charge on microplastics, the hydrolysis mode of a coagulant and other factors [77]. The pH value determines the type of hydrolysis taking place when using inorganic coagulants [74]. An increase in the pH value causes the negative charge on the particles’ surface to become greater [78]. pH also affects the particle size of the flocs, with larger floc sizes for alkaline conditions than acidic conditions [79]. At low temperatures, the movement and collision energy of the particles are low, resulting in fewer collisions between particles. This results in weak floc aggregation [80].

The types and dosages of coagulants also substantially influence the efficiency of coagulation. The dose of the coagulant used has a substantial effect on microplastic removal. The relation between the efficiency and dosage of coagulation depends on the primary mechanism of coagulation. If the primary mechanism is simple charge neutralization, the removal efficiency will generally increase with the coagulant dosage. This is because the absolute zeta potential value of the microplastics will gradually decrease with the addition of a coagulant. Maximum removal is achieved when the zeta potential of the microplastic is 0 [67]. If the coagulation process occurs due to several coexisting mechanisms, then microplastic’s relative stabilization phenomenon does not occur if the coagulant dose is too high [68]. For the sweeping mechanism, it is assumed that a large dose of a coagulant will cause the density and structure of the flocks to be greater with stronger adsorption and sweeping effects [81]. The types of coagulants typically used are categorized into two: chemical coagulants, which include inorganic coagulants, organic synthetic coagulants and polymeric coagulants, and natural coagulants [74]. Most current research and practice on microplastic removal coagulation processes use inorganic chemical coagulants. However, in recent years, natural biological coagulants have gained importance. Inorganic coagulants include small-molecule inorganic coagulants such as aluminum trichloride and aluminum sulfate and inorganic polymeric coagulants (IPCs) such as PAC and PFS. IPCs have a higher charge density and molecular weight compared to small-molecule inorganic coagulants [82]. Aluminum-based coagulants are largely found to be more efficient in removing microplastics than other chemical coagulants. Inorganic coagulants are the most commonly used coagulants in the industry. Organic polymer coagulants consist of long-chain polymers, which can be classified based on their ionic disposition. These polymers, when hydrolyzed, can stimulate particle aggregation through bridging or charge neutralization [83]. The polymer coagulants commonly used to remove microplastics are PAM, polyamines, and others. These coagulants can be inserted into water bodies to remove microplastics [68]. Natural coagulants have recently been studied as a substitute for chemical coagulants. Natural coagulants are cost-effective due to their abundance in nature. The dosage of natural coagulants required is usually lower, and the coagulants are stable. Another advantage natural coagulants have over chemical coagulants is that they pose a lower toxicological risk due to their formation from nature [84]. These coagulants primarily originate from organic polysaccharide materials such as chitosan and starch. The macromolecular form and additional functional groups of the natural coagulants help with the neutralization of the negatively charged microplastics [85].

It is important to note the characteristics, such as the size and shape, of the microplastics to be removed, as they play an important role in coagulation. Although the characteristics are not a key factor, different microplastic removal studies have observed a considerable difference. The size of the plastic influences the rate of collision and settling behavior [76]. It is difficult with current research to know how the particle size relates to removal efficiency as there have been studies concluding that smaller and larger particle sizes are easier to remove. As for the morphology of plastic particles, it has been consistently found that fibers are best removed by coagulation [71,86,87].

Other factors affecting the removal efficiency of coagulation include turbidity, water flow rate, stirring intensity, and flocculation and sedimentation time. The coagulant dosage required is lesser at high turbidity as the collision frequency is higher [88]. Li et al. found that shorter sedimentation and coagulation time caused the efficiency of removing microplastics to be low. The study also concluded that there was a low efficiency at high stirring speeds due to a reduction in the particle size in flocs [89].

### 3.3. Chemical Coagulants

Chemical coagulants, which include inorganic coagulants and organic synthetic polymer coagulants, are efficient in removing microplastics in wastewater systems. Inorganic coagulants have a strong reaction with the negatively charged microplastics through the cations produced by hydrolysis [90]. In general, aluminum-based coagulants have better efficiency in removing microplastics than iron-based coagulants. Table 1 summarizes a few recent studies on removing microplastics using chemical coagulants in wastewater systems.

Al-based and Fe-based inorganic coagulants are the most commonly used coagulants in the research conducted on microplastic removal in wastewater. In recent years, Mg-based coagulants have been used in certain studies and have shown promising potential. Factors such as pH, the dosage of water and the presence of other substances in the sample water affect the efficiency of these coagulants. In summary, Fe-based coagulants are more effective in removing microplastics than others. However, the size and type of microplastics present in the water also play a major role in the coagulation process, and further research needs to be conducted. An increase in the coagulant dosage does not necessitate higher removal efficiency. It can be observed that increasing the coagulant dosage past the optimum value tends to decrease the removal efficiency. It is difficult to determine the effect of pH on the efficiency as it depends on the microplastic present and the coagulant used.

### 3.4. Natural Coagulants

Natural coagulants have a cost and environmental benefit over chemical coagulants. In recent years, natural coagulants have gained much importance in scientific communities as chemical coagulants are found to be toxic to the environment. Chemical coagulants are not biodegradable and tend to persist in water unless treated specifically [98]. The presence of aluminum, one of the most commonly used chemical coagulants, in drinking water has been linked to contributing to Alzheimer’s and related diseases in humans [99]. Natural coagulants derived from plant sources can overcome these health concerns. In addition to the health issues, chemical coagulants pose a threat to the environment as they produce hazardous sludge. Natural coagulants, on the other hand, do not increase the metal load, and they produce minimal waste sludge, making them a sustainable alternative [98]. The dosage of natural coagulants required is also lower than that of their chemical counterparts, making them cost-effective. Considering all these advantages, natural coagulants are a far superior and sustainable alternative to chemical coagulants. Limited research is found on the efficiency of natural coagulants, with most research focusing on turbidity and COD removal. However, limited studies have proven natural coagulants as a worthwhile substance for removing microplastics. Although the number of studies is limited, green coagulants have exhibited a promising future in the wastewater industry by effectively removing turbidity, COD, BOD and TSS from wastewater. There is a lack of research on the industrial-scale application of natural coagulants. In addition, there is a need for more research on the optimization of parameters for natural coagulants. With proper scale-up and optimization, natural coagulants can replace chemical ones in WWTPs. Table 2 summarizes the recent research conducted on the removal of microplastics using natural coagulants.

Natural coagulants play a substantial role in removal efficiency when used in water treatment. Most natural coagulants used and summarized in the table above exhibit high removal efficiency. The studies summarized in the table above show that a higher coagulant dosage was directly related to a higher removal efficiency. However, this varies with the coagulant used, and further research needs to be conducted before a conclusion can be made. The active coagulant agents, polysaccharides, proteins and polypeptides also need to be studied as they play an essential role in the efficiency of these coagulants. It is observed from most studies that natural coagulants work best at an optimum pH of 7 with slight variations.

### 3.5. Use of Natural Coagulants in Southeast Asia

Coagulation is a common process involved in most wastewater treatment plants. Most plants use chemical coagulants, which have toxic effects on the environment and are also expensive. As cost is one of the major barriers to wastewater management in Southeast Asia, the coagulation process with the use of natural coagulants is a suitable alternative. There currently needs to be more research available on the removal of microplastics by coagulation in WWTPs in Southeast Asia. Furthermore, no studies have been conducted on natural coagulants in this region. However, in numerous studies conducted worldwide, coagulation achieved excellent results in microplastic removal. The natural coagulants researched also showed the potential to replace chemical coagulants in the industry for treating wastewater.

*Diascora hispida* is a plant found in the tropic and subtropic regions of the world, especially in West Africa, the Caribbean, and Southeast Asia [110]. *Diascora hispida* was used as a natural coagulant to treat textile wastewater effluent and could achieve an efficiency of 28%, 94% and 64% at an optimum pH [111]. Banana has the highest production amount in Southeast Asia [112], and a study conducted on banana peels as a natural coagulant showed the removal of 88% of turbidity under optimum conditions [16]. One study conducted in Malaysia used the local plants *A. indica*, *S. palustris*, *D. linearis*, *S. polyanthum*, *M. esculenta*, *P. sarmentosum* and *M. malabathricum* to effectively remove turbidity, with *P. sarmentosum* having the highest efficiency of 24.2% removal at a small dose of 5 g/L [113].

Moringa, a natural coagulant, has been extensively researched by scientists and has shown promising coagulation properties. Improving the extraction method for this coagulant can immensely enhance its coagulant activity. In one study, extraction of the moringa plant using salt solutions had a removal efficiency of 91%, which was significantly higher than that when the coagulant was extracted with water [114]. In addition to improving the extraction, purifying the natural coagulants has also been linked with greater efficiency. One of the ways of purifying the coagulants is by lipid removal [115]. The coagulant activity depends on the active compounds, polysaccharides and proteins; thus, maximizing the extraction of these compounds will ensure higher removal efficiency. Other plant-based coagulants such as *Ocimum basilicum* and hibiscus, which are native to the Southeast Asian regions, have shown great potential [116]. *O. basilicum* reduced COD by 61.6% and dye by 68.5% at a low dosage of 1.6 mg/L [117]. These natural coagulants show promising effectiveness and can be further studied by enhancing their extraction and purification methods so they can be used in WWTPs. Furthermore, strategies such as hybrid processes and modifications to the natural coagulants are also worth mentioning when improving efficiency. Hybridizing involves composite coagulants, where the natural coagulants are chemically modified with inorganic coagulants. This process not only enhances the efficiency of the coagulation process, but also reduces the harmful impacts of the chemical coagulants due to the presence of the natural substances [118].

The removal efficiency of natural coagulants shows promising effects for removing microplastics from WWTPs. Even though the efficiency of chemical coagulants is also high, the substantial impact of chemical coagulants on the environment and living beings is a major drawback. Natural coagulants are cheap and non-toxic, making them a suitable alternative for mitigating microplastics released into the environment from WWTPs. Many natural coagulants that can be used to remove microplastics from wastewater influents are abundant in Southeast Asia.

## 4. Future Perspectives

Further research on the characteristics of microplastics present in wastewater treatment plants in Southeast Asia is required. The relation between the characteristics of microplastics and coagulation efficiency needs to be studied in further detail. Experiments conducted using natural coagulants to remove microplastics will also help with the implementation of these techniques in the future. It is also worth looking into the extraction and purification of natural coagulants to ensure they are able to operate at the maximum possible capacity.

## 5. Conclusions

The key findings of this review include the following: (i) Billions of tons of microplastic are present in the marine environment, with the majority coming from land sources. Wastewater treatment plants in Asia, particularly Southeast Asia, are ineffective in removing microplastics. The discharge amounts to more than half of the marine plastic waste. (ii) Wastewater treatment plants can remove a certain amount of microplastics during treatment. However, the plants need to be equipped with specific treatment technologies. Implementing new technologies is costly; thus, the optimization of current processes is a better alternative. (iii) The optimization of the coagulation process could help mitigate the microplastic problem in WWTPs. Natural coagulants are cheaper and more sustainable than chemical coagulants. (iv) The abundance of natural materials in Southeast Asia represents a potential for the region to implement natural coagulation in WWTPs to diminish the microplastic problem.

## Figures and Tables

**Figure 1 toxics-12-00012-f001:**
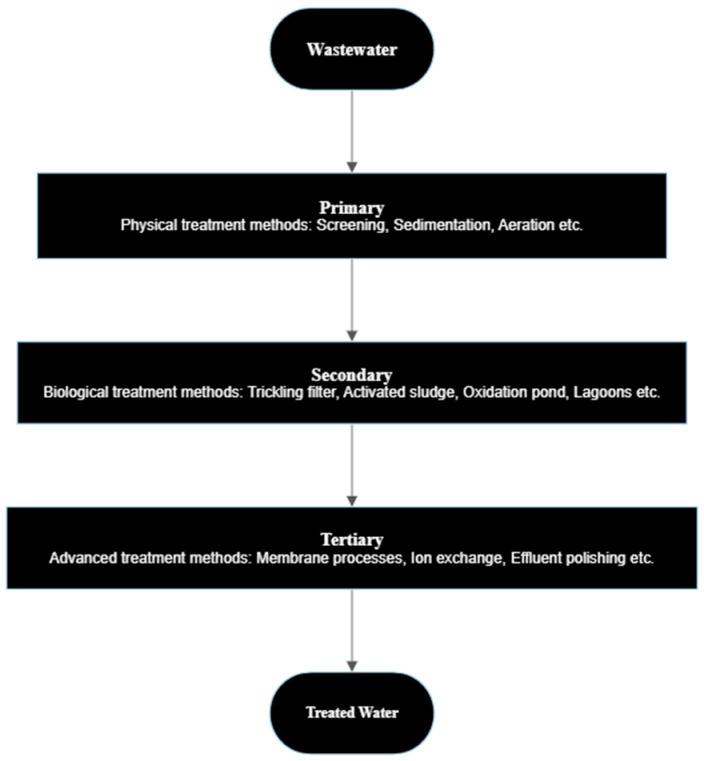
Classification and technologies in different steps of a wastewater treatment plant.

**Figure 2 toxics-12-00012-f002:**
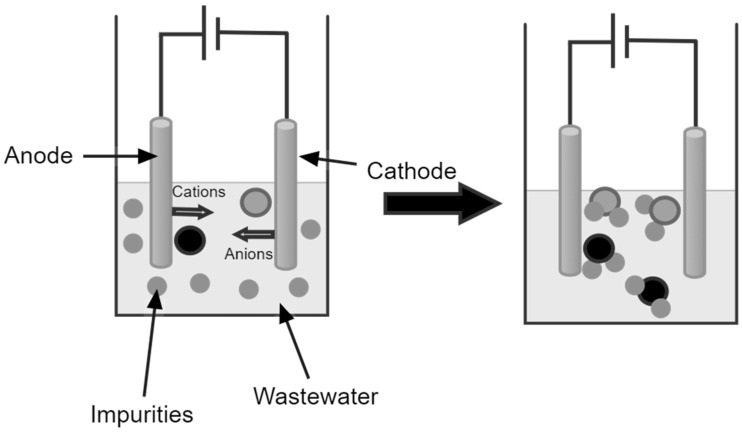
Electrocoagulation process.

**Figure 3 toxics-12-00012-f003:**
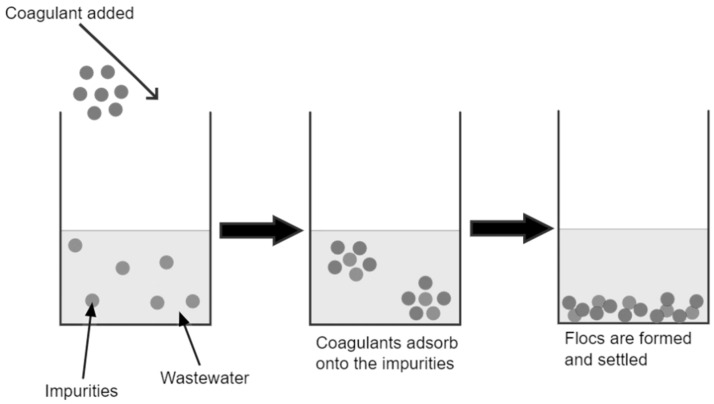
Coagulation process.

**Figure 4 toxics-12-00012-f004:**
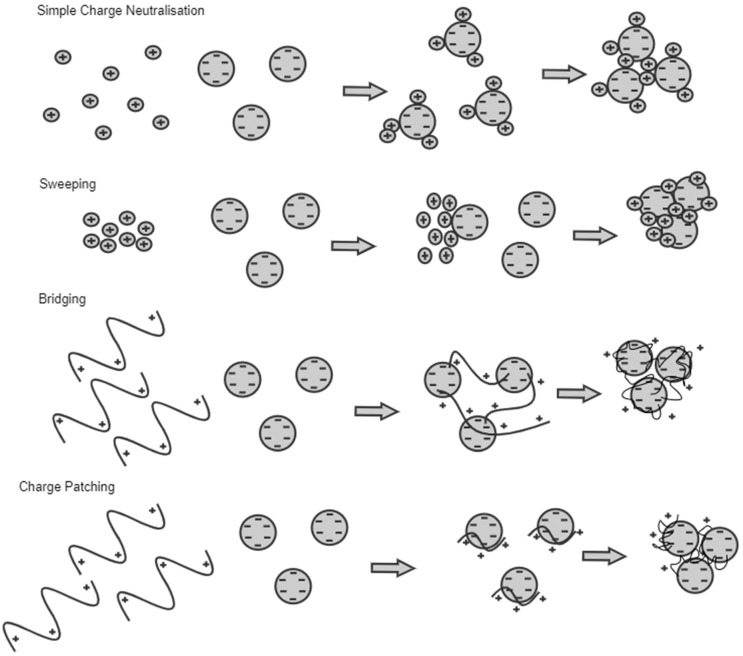
Mechanism of the coagulation process.

**Table 1 toxics-12-00012-t001:** Results of microplastic removal using chemical coagulants.

Location	Coagulant	Dosage of Coagulant	Sample	pH	Efficiency	Reference
Ontario, Canada	Aluminum hydroxide	40 mg/L	River water	7.8	71%	[91]
Czech Republic	Alum	-	Drinking water treatment plant	3.5	61.65%	[92]
Surabaya City, Indonesia	Aluminum sulfate (Al_2_(SO_4_)_3_)	-	River water	-	17%	[37]
Daegu, Republic of Korea	PAC	WWTP A—32.4 mg/L WWTP B—30.5 mg/L WWTP C—29.3 mg/L	Wastewater treatment plant	-	WWTP A—53.8% WWTP B—81.6% WWTP C—47.1%	[69]
Tianjin, China	PAC, PAM, Fe_3_O_4_	-	Constructed wetland	-	73.8% (sunny days) 77.9% (rainy days)	[70]
Detroit, MI, USA	Aluminum sulfate	20 ppm	Water treatment plant	7.43–7.59	13.6% (particle size 45–53 µm)	[93]
Australia	Alum, PAM	50–250 mg/L	Simulated stormwater	3–11	Maximum: 96% at 150 mg/L alum and 15 mg/L PAM	[94]
Finland	Ferric chloride, polyaluminum chloride	0.017–1.4 mmol/L	Wastewater	6.5	Ferric chloride—99.4% Polyaluminum chloride—98.2%	[81]
China	Magnetic magnesium hydroxide Mg(OH)_2_, iron oxide (Fe_3_O_4_)	200 mg/L Mg(OH)_2_ 120 mg/L Fe_3_O_4_	Simulated wastewater	7	66.3 to 87.1%	[95]
-	Iron (III) chloride (FeCl_3_) PAC	30 to 180 mg/L (30 increments)	Simulated water	7	PS—77.83% PE—29.70%	[79]
China	Aluminum chloride (AlCl_3_), calcium chloride (CaCl_2_)	-	Lake water	3–10	More than 80% at pH > 6	[96]
Greece	Iron sulfate (FeSO_4_), iron (III) chloride (FeCl_3_), magnesium sulfate (MgSO_4_)	496–993 mg/L FeSO_4_ 483–964 mg/L FeCl_3_ 1025–2050 mg/L MgSO_4_	Tap water	8	92.4% for Fe^2+^ ion 89.1–90.4 for Mg^2+^ ion	[97]

**Table 2 toxics-12-00012-t002:** Water treatment using natural coagulants.

Coagulant	Dosage of Coagulant	Sample	pH	Efficiency	Reference
Pinecone extract	0.5 mL/L	Synthetic turbid water	2 and 12	Maximum turbidity removal: 82%	[100]
*Salvia hispanica* (chia)	40 g/L	Landfill leachate	7	Turbidity: 62.4% COD: 39.76%	[101]
*Strychnos potatorum*	40.0 mg/L	Artificial water	7	Kaolinite turbidity: 93%	[102]
*Leucaena leucocephala*	10 mL/L	Synthetic wastewater	3	Congo red dye: 99.9%	[103]
Cactus (*Opuntia ficus-indica*)	1500 mg/L	Oil sand process-affected water	7–8	Turbidity: 98%	[104]
Rice husk ash	6.0 g	Palm oil mill effluent	3.6	COD: 52.38% TS: 83.88%	[105]
*Moringa oleifera*	50 mg/L	Surface water	7.03–7.70	Turbidity: 85%	[17]
*Phaseolus vulgaris*	0.5 M	Synthetic turbid water	7.4	Turbidity: 85%	[106]
Fava bean seeds (*Vicia fava* L.)	0.125–0.25 mL/L	Synthetic water	10	Turbidity: 51.5 to 54%	[107]
*Musa paradisica* (banana) peels	0.6 mL/L	Simulated turbid water	11	Turbidity: 98.14%	[108]
*Dolichos lablab* (Indian beans)	0.6 mL/L	Simulated turbid water	11	Turbidity: 98.84%	[108]
Soybean	120 mg/L	Surface water	-	Turbidity: 23.2% Color removal: 30.4%	[109]

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
