# Peer review of "Microplastic Removal in Wastewater Treatment Plants (WWTPs) by Natural Coagulation: A Literature Review"

_toxics, 2023, doi:10.3390/toxics12010012_

Round 1
Reviewer 1 Report
Comments and Suggestions for Authors
This manuscript provides a review of the current status of wastewater treatment plants in Southeast Asia regarding the treatment of microplastics in influent and the effectiveness of coagulants in microplastic treatment. This review is important for understanding the potential use of natural coagulant in removing microplastic. However, as specified below, this manuscript needs significant improvement in many ways.
1. The article dedicates significant sections to elucidate the sources of microplastics in Southeast Asia, the hazards of microplastics, and wastewater treatment processes in the region. However, there is comparatively less information regarding the effectiveness of coagulation in removing microplastics. It is necessary to restructure the article and allocate more space for a detailed explanation of the coagulation's efficacy in microplastic removal.
2. Line 106: It is necessary to introduce the physical and chemical properties of microplastics, such as the functional groups contained, hydrophilicity and so on.
3. Line 232: In the drinking water treatment process, coagulation processes are beneficial for removing larger microplastic particles. It is important to provide information regarding the size and primary components of these microplastics found in the source water.
4. Line 303: The introduction mentions that primary and secondary treatment can remove 66% of microplastics, which appears contradictory to this point. Further clarification is needed.
5. Line 331: In both scenarios of adding and not adding coagulants, the removal efficiency of conventional primary treatment for microplastics needs to be separately elucidated.
6. Line 345: A brief overview of the treatment processes used in different wastewater treatment plants in various regions is required.
7. Line 352-353: The statement here is not precise. After tertiary treatment, the overall removal rate of microplastics can reach 99.9%, and it can be considered that tertiary treatment achieves a removal rate close to 100%, significantly higher than that of primary and secondary treatment.
8. Line 399: Missing abbreviation labeling.
9. Line 394: An introduction to the treatment processes of CCT and CWT is needed.
10. Line 446: This is the mechanism of coagulation. Further elaboration is required to explain the coagulation mechanism for microplastic removal.
11. Line 532-534: A summary and in-depth analysis of the content in the table are required.
12. Line 557-558: A summary and in-depth analysis of the content in the table are required.
13. Line 579: This section describes the removal efficiency of natural coagulants on turbidity, not on microplastics, which is unrelated to the main theme of this article. Furthermore, at a dosage of 5g/L, the removal efficiency is only 24.2%, which does not demonstrate any advantage over chemical coagulants.
Reviewer 2 Report
Comments and Suggestions for Authors
The authors present an interesting topic on microplastics and their methods are discussed. However, I consider that there are important references that have not been cited from the literature.
Here an example
https://doi.org/10.1016/j.envres.2022.114224
The figures or diagrams must be improved, they are low resolution.
The authors titled their work "Microplastic Removal in WWTPs by Natural Coagulation: A Study in Southeast Asia", they must adjust the content of their work only to the coagulation processes and delve into the use of flocculating coagulants, advantages and disadvantages, etc.
His work cites other microplastic removal technologies that do not agree with the title and abstract.
Additional comments:
In my report I indicate that the title is not appropriate for the content of the manuscript.
The work contains general information on different technologies for the removal of microplastics and the title only refers to the use of coagulants.
The authors should deepen the use of natural coagulants for the removal of microplastics. They must list in a table the different natural coagulants, performance parameters and type of microplastics studied.
The literature must be updated, there are many important references that are not mentioned and discussed.
The work could be considered in another journal, for example separation.The content of the manuscript does not refer to the toxic impact of microplastics on humans and the environment.
Reviewer 3 Report
Comments and Suggestions for Authors
The article provides an overview of the pervasive issue of microplastic pollution caused by urban industrialization, particularly in Southeast Asia. It emphasizes the growing threat to aquatic and human life due to the bioaccumulation and biomagnification of plastics. The major contribution of wastewater treatment plants (WWTPs) to this problem is highlighted, underscoring the need for improved microplastic removal technologies.
The mention of coagulation as a significant process for microplastic removal and the superiority of natural coagulants over chemical alternatives due to their non-toxicity and cost-effectiveness is relevant and promising. However, the article would benefit from a more explicit emphasis on the novel aspects or innovative approaches in this context. This could include highlighting specific natural coagulants that have shown remarkable effectiveness, presenting unique strategies for their application in WWTPs, or discussing recent breakthroughs and developments in this field.
While the article effectively addresses the critical issue of microplastic pollution in Southeast Asia and the potential of natural coagulants in WWTPs, it would be greatly strengthened by accentuating the novelty and innovation in this field as there is a number of works dealing with this subject.
Comments on the Quality of English LanguageMinor editing of the English language is required.
Reviewer 4 Report
Comments and Suggestions for Authors
This is a very long review article which should be, in theory, strictly focused on an interesting topic, i.e. the microplastic removal in WWTPs by natural coagulation. The topic of this review is interesting, however, the article needs to be fully revised before publication. The introductory section is too long, generic and dispersive. The first two chapters focus on very broad and wide general topics in an anecdotal way. The information presented in these two sections is not novel nor interesting and it could be removed. So, i warmly suggest the authors remove entirely sections 1 and 2 from their manuscript as these sections do not add anything to the general topic of this review. Moreover, the information contained in these two chapters is often erroneous, too generic and misleading. This review article can be greatly shortened and the article can start directly from section 3 without detriment to the overall quality and interest of this manuscript. In this way, this review article can be greatly sharpened and strictly focused on the very specific topic of this review.
Below are minor additional review details:
- Lines 122 and 126: the same concept is repeated twice. Please remove repetition. In line 122 the word "size" should be replaced by "shape" right? Anyway, as i said above, i would entirely remove sections 1 and 2 from this manuscript as they are really not relevant.
- Line 149: Thompson et al. was not the first one to provide evidence of microplastic contamination in the marine environment. Several other authors provided similar evidence decades earlier than that paper was published. Please remove (or better, entirely remove sections 1 and 2).
- Line 287: remove repetition of the word "fragments" on this line.
- Line 398: replace "CCT" with "CWT" right?
English language is fine, but it could be improved.
Round 2
Reviewer 1 Report
Comments and Suggestions for Authors
Most of the issues previously raised in this text have been addressed. However, if the following issues cannot be resolved, the article cannot be accepted.
1. Sections 5.4 and 5.5: This section elaborates on the application of natural coagulants in wastewater treatment rather than focusing on the removal effectiveness of microplastics. The cited studies should be relevant to the removal of microplastics using natural coagulants.
Author Response
all comments are been addressed accordingly

Reviewer 4 Report
Comments and Suggestions for Authors
The authors did not address the main comment I made during the first revision and Sections 2 and 3 have not been removed as I suggested during the first revision. As already said, the introductory section is too long and dispersive. The first two chapters focus on very broad and wide general topics in an anecdotal way. The information presented in these two sections is not novel nor interesting and it could be removed. So, i warmly suggest the authors to remove entirely sections 1 and 2 from their manuscript as these sections are not adding anything to the general topic of this review. The information contained in these two chapters is often erroneous, too generic and misleading and most of all is not relevant to the main aim of this review. This review article can be greatly shortened, and it can start directly from section 3 without detriment to the overall quality of the manuscript.
Title: I suggest to remove “A Study in Southeast Asia” from the title of this manuscript. This is not “a study in southeast Asia”, it is a generic review on a very specific issue, i.e. Microplastic Removal in WWTPs by Natural Coagulation. Perhaps, the authors can replace “a study in southeast Asia” with “a literature review”. By doing so, the title of this manuscript would become: “Microplastic Removal in WWTPs by Natural Coagulation: a literature review”.
Lines 97-98: the aim (1) “review the sources of marine microplastic pollution, especially in Southeast Asian countries”, can be removed from this list as this aim is not relevant to main goal and the most interesting aims of this review and it is also too broad and not properly addressed here. I suggest the authors to remove aim (1) and strictly focus on the other 3 aims which are less broad and generic, and much more focused and interesting, as it is also reflected from the title of this review.
Table 1, Fig. 1, Fig. 2, Fig. 3, Fig. 4 and Fig. 5 are not relevant to the main aims of this review, these are generic anecdotal figures, they do not deal with microplastic removal in WWTP by coagulation, and so they can be removed without detriment to the overall quality of this review. They do not add anything to this manuscript. Again, I suggest the authors to strictly focus on what is stated in the title of this review, i.e. Microplastic Removal in WWTPs by Natural Coagulation.
Lines 148-149: Again, Thompson et al. was not the first to propose the term microplastic. The term was first introduced in the scientific literature by Ryan, P.G. and Moloney, C.L., 1990. Plastic and other artefacts on South African beaches: Temporal trends in abundance and composition. S. Afr. J. Sci./S.-Afr. Tydskr. Wet., 86(7), pp.450-452. https://www.researchgate.net/publication/283507743_plastic_and_other_artefacts_on_South_African_beaches_temporal_trends_in_abundance_and_composition
Comments on the Quality of English Language
Minor editing of English language required
Author Response
all comments from reviewer 4 addressed accordingly

Round 3
Reviewer 4 Report
Comments and Suggestions for Authors
Thanks for addressing my comments.
Comments on the Quality of English LanguageModerate english language revision is needed.